# The Mentor Mothers Program in the Department of Defense in Nigeria: An Evaluation of Healthcare Workers, Mentor Mothers, and Patients’ Experiences

**DOI:** 10.3390/healthcare9030328

**Published:** 2021-03-14

**Authors:** Josephine Moshe Ibu, Euphemia Mbali Mhlongo

**Affiliations:** School of Nursing and Public Health, University of KwaZulu-Natal, Durban 4041, South Africa; mhlongoem@ukzn.ac.za

**Keywords:** Mentor Mothers program, experiences, patients, healthcare workers, vertical transmission, PMTCT care

## Abstract

Nigeria contributes the highest to the global burden of HIV/AIDS and also accounts for the largest proportion of new vertically transmitted HIV infections among children. The Mentor Mothers program in the Nigerian Department of Defense was introduced in accordance with the World Health Organization and its implementing partner guidelines to curb the high incidence of vertically acquired HIV infections. Understanding the experiences of participants could serve as a gateway to evaluating the effectiveness of the program to better provide quality services within targeted health facilities. This qualitative study employed key informant interviews with six healthcare workers as well as two focus group discussions with six mentor mothers and six prevention of mother-to-child transmission (PMTCT) patients in four selected hospitals in the Nigerian Department of Defense to explore their experiences of the Mentor Mothers program. A thematic analysis technique was used to analyze the collated data. As a result, four main themes emerged, with the program perceived by most participants as providing psychosocial support to the patients, a valuable educational resource for raising HIV awareness, a valuable resource for promoting exclusive breastfeeding and mitigating vertical transmission of the virus, and functioning as a link between patients and the healthcare system. The participants reported that the program had effectively decreased HIV infections in children, reduced child and maternal mortality, and supported the livelihood and development of women, families, and communities in and around the Nigerian Department of Defense health facilities.

## 1. Background

In 2017, the Joint United Nations Program on HIV/AIDS (UNAIDS) report revealed that out of the 180,000 new infections among children globally, about 159,000 occurred in sub-Saharan Africa, with Nigeria contributing the largest proportion of about 23% of new infections within the sub-Saharan region [1]. For example, in 2009, 21 out of the 22 countries with the highest prevalence of women living with HIV were recorded in Africa, which also accounted for about 90% of pregnant women living with HIV needing services for prevention of mother-to-child transmission (PMTCT) of HIV [2]. Recent studies conducted in Nigeria have revealed that despite improved access to antiretrovirals (ARVs), the utilization of PMTCT services is below the global standard. Two reports conducted by UNAIDS in 2012 revealed that only about 62% of HIV-infected pregnant women in Nigeria received ARVs for PMTCT [3,4], and only 35% of infants born to HIV-infected mothers in Nigeria were tested within the first two months of delivery [3]. Although the prevalence of children born with HIV infection has declined by 58% since 2002 in Nigeria, about 240,000 infants were born with HIV in 2013 in the country [5,6], which is above the global standard of 20,000. Furthermore, in 2017, about 160,000 HIV-infected pregnant women in Nigeria required ARV drugs, making it the second highest number globally after South Africa [7]. The first PMTCT of HIV program commenced in the country in 2002 [8]. Since then, ARV coverage for PMTCT in the country has been very poor [9]. For example, a study revealed that only 32% of pregnant women with HIV received ARV drugs in 2017 for PMTCT care [10]. This is an indication that limited coverage of PMTCT care has continued to plague Nigeria’s performance in curbing new vertical transmission of HIV among children.

Given the introduction of global targets and programs to end vertical transmissions of HIV among children and the current global aim to end the HIV epidemic by 2030 [11,12,13,14], understanding the experiences of healthcare providers and patients who receive care could serve as a gateway to evaluating the effectiveness of the program and how to better provide quality services within targeted health facilities. To achieve these on a global scale, each of these necessary ambitious goals and strategies require a well-trained healthcare workforce and customized frameworks. Having a well-structured and well-functioning health system that cultivates quality service delivery, including ensuring skilled healthcare providers, availability of basic and essential HIV amenities, and adequate funding, is vital to achieve the desired impact of PMTCT programs globally [15,16,17].

As HIV-positive women who have gone through PMTCT services and are trained to mentor and support other HIV-positive mothers, mentor mothers (MMs) in the Mentor Mothers program of the Nigerian Department of Defense (DoD) provide services that range from HIV prevention, health education, counselling and promotion of HIV testing, tracking lost to follow-up patients, raising awareness, and community involvement to providing adherence counselling. These MMs offer these services through individual counselling as well as other activities in the community and health facilities where they work within the Nigerian DoD health facilities. However, studies that measure or evaluate healthcare workers, mentor mothers, and patients’ experiences of the MMs program targeting PMTCT care remain limited. Two studies assessing mentorship programs that specifically target PMTCT care suggested a reduction in vertical transmission [18,19], but such studies are few and further investigation is required on the phenomenon [20]. One retrospective study of HIV-positive women also revealed that support group participation targeting PMTCT practices was associated with increased longevity among patients due to the positive role that these support groups play in patient retention [21].

## 2. Aim

The aim of this study was to evaluate and present an analysis of the experiences of healthcare workers, mentor mothers, and patients in the Mentor Mothers program targeting PMTCT practices within the Nigerian DoD. The results of this study will not only be instrumental in improving the effectiveness of services rendered within the Nigerian DoD health facilities but will also form part of suggestions and resolutions to aid the development of a framework to facilitate implementation of the program in the rest of the health facilities that are not utilizing MMs program within the Nigerian DoD.

## 3. Research Methods

### 3.1. Research Design

A qualitative descriptive study was conducted to explore the experiences of participants in the MMs program in the Nigerian DoD. This method was deemed appropriate because it is particularly valuable in examining healthcare and other issues related to nursing as it allows for in-depth interviews, questioning, and further probing of participants based on their responses in order to enable interviewers/researchers to understand their thoughts. Moreover, this study focused on gaining insights from categories of participants regarding their experience in the MMs program. To this end, in-depth interviews and focus group discussions were conducted to explore these experiences. As critical stakeholders contributing to PMTCT care and services, this approach is adequate to describe their experiences.

### 3.2. Research Setting

The research was carried out in four selected health facilities of the three-armed forces of the Nigerian Army, Navy, and Airforce within Abuja, the Federal Capital Territory (FCT) where PMTCT care and services are provided. With an estimated total population of 3,277,740 in 2020, the FCT is home to 12 military barracks and health facilities. Although they all offer PMTCT services, only six of the 12 health facilities offer the MMs program’s services targeting PMTCT. These include the Guards Brigade Medical Centre, Mambila Barracks; 063 NAF Base Hospital, Airport Road Abuja; and the Defense Headquarters (DHQ) Medical Centre, Abuja. The others are the Lungi Barracks, Gado Nasko Barracks, and Aguyi Ironsi Barracks. The DHQ Medical Centre serves as the coordinating unit for all the Nigerian DoD MMs program in Nigeria. The choice of the selected facilities in Abuja was strategic as Abuja is the administrative headquarters of the Nigerian military medical corps, where the Nigerian DoD MMs program originated in 2014. Although these services were primarily targeted toward military personnel and their families, MMs services are also provided to nonmilitary personnel and their families. 

### 3.3. Research Participants

A purposeful sampling method was used to recruit participants, consisting of six key informants made up of three doctors and three nurses trained for PMTCT across the selected hospitals, six mentor mothers, and six PMTCT patients. These participants, especially the doctors and nurses, were purposely chosen by virtue of their training and vast knowledge in PMTCT practices. Participants were selected if they had been in the program for a minimum of six months. The inclusion criteria for the patients were that they were confirmed seropositive pregnant women registered and attending antenatal care (ANC)/PMTCT clinics in the selected health facilities with a minimum of four clinic visits and that they had been in the program for a minimum of six months. The table below (Table 1) shows the participants profile and duration of engagement in the Nigeria DoD mentor mother program.

### 3.4. Recruitment Process

Nurse managers and hospital administrators were contacted for introduction of the research topic and to gain access to participants. Participants that met the criteria were purposely selected based on their direct involvement in the MMs program in the DoD health facilities. Written and verbal consent to voluntarily participate in the study was obtained from each participant.

### 3.5. Instrument

The research instrument was developed by the researcher after conducting literature review. This process was guided by the World Health Organization (WHO) health system responsiveness key informant questionnaire, key informant survey, and the health system performance assessment (HSPA) tool [22,23,24]. Appendix A provides detailed information on the types of questions included in the guide.

## 4. Ethical Considerations

Ethical approval was obtained from the University of KwaZulu-Natal (UKZN) ethical committee (protocol number 00000186/2009) and the Ministry of Defense Health Research Ethics Committee (MODHREC) (reference number MOD/HIP/G3/230/09). Standard protocols as stipulated in the Helsinki declaration to ensure confidentiality and anonymity were also adhered to throughout the data collection and analysis processes.

### 4.1. Data Collection

Interviews of the six key informants were conducted with doctors and nurses, while two focus group discussions were conducted with the MMs and PMTCT patients separately. The interviews were conducted in English language, lasted between 30 and 45 min each, and were sound recorded with the permission of participants. Data was collected between 5 and 14 February 2020.

### 4.2. Data Analysis

The audio recordings were transcribed verbatim and analyzed using a thematic analysis (TA) approach as described by Clarke and Braun and Rice and Ezzy [25,26]. The analysis was a continuous process in which the collated data were repeatedly and independently read multiple times by the researchers to ensure that the constructions of the concepts were fully grasped and documented by the researchers. This process also allowed the researchers to get familiar with the different responses from the participants and the ideas and meanings attached to their constructions. Thereafter, the researchers identified common constructions or meanings that emerged from the coded responses. These were then deconstructed to gain a better understanding of the discourses and categorized as themes. Contradictions, similarities, and ambiguities emerging from the responses were deconstructed based on how the participants discussed the different themes that emerged from the dataset. The researchers also focused on the historical background as well as the context within which the participants were operating to gain a better insight of the themes that emerged. Thereafter, the draft of the data analysis and the coded collated data was sent to a research analyst to review the presentation of the results. Comments and suggestions were recommended by the analyst, which were taken into consideration by the researchers. Consequently, the researchers constructed a holistic and comprehensive view of the emerging discourses or constructions through an iterative process. The codes were reviewed by the researchers and the analyst, and the results were compared with similar studies from other settings. The participants were assigned pseudonyms to ensure they were not linked to the data collected. To properly evaluate the experiences of the participants, the TA technique helped to illustrate the relationships between key themes and their impact on the MMs program within the Nigerian DoD health facilities. Here, the researchers summarized the results under each theme in a narrative style that described the participants’ experiences of PMTCT through the MMs program while still relating the findings to the collected data.

### 4.3. Trustworthiness

This was achieved by coding and comparing coded data and using thick descriptive data to define resultant themes, utilizing the purposive sampling strategy that also ensured an adequate spread of participants from each of the military health facilities selected for study, and describing the research processes in detail to enable its replication in other settings as required. Being part of a larger research project, the supervisor of the project provided oversight and ensured its dependability. Confirmability was ensured by rechecking the emergent themes generated from the data, using thick descriptive data to substantiate emergent themes, and comparing the data with previous research findings. In addition, the findings from this study were checked with the participants in a follow-up meeting to ensure that the results presented in this paper are a true reflection of the participants’ views. This also improved the trustworthiness of this study. 

Four main themes emerged from the dataset, as shown in Figure 1 below. The MMs program in the DoD was perceived by most participants as providing psychosocial support to the patients, a valuable educational resource for raising HIV awareness, a valuable resource in promoting exclusive breastfeeding and mitigating vertical transmission of the virus, and functioning as a link between patients and the healthcare system. 

## 5. Psychosocial Support

Most of the participants perceived receiving and/or giving a diagnosis of HIV to be a life-altering experience for them because being HIV-positive is mostly associated with infidelity, promiscuity, and/or witchcraft, as reported by most of the participants. As a result, MMs were highly perceived to provide psychosocial support to other HIV-positive women, which helped in reducing the stigma and sense of alienation that these patients were initially faced with. Other participants associated their experiences with the MMs as motivating, encouraging, loving, selfless, compassionate, trusting, and giving them a sense of hope for improvement. As HIV-positive mothers, the MMs in the program understand and have deep empathy for the women they render their services to. They embody the Nigerian DoD MMs program goals and believe that positive change is possible in the country’s poorest communities. They know that if the right information, support, and tools are given, women have the strength within themselves to turn their lives around positively.

The narratives below highlight the views of some of the MMs:


*“It’s all about trust … they really trust us, they believe whatever we tell them. Most of their husbands trust me because I am working with the knowledge I have … With the help of mentor mothers, these women can boldly breastfeed their babies even in the market, in the church, anywhere or even in front of family members who know they are HIV-positive”*
(mentor mother)


*“They believe us more than they do the nurses”*
(mentor mother)

On their part, some of the patients related their experiences while appreciating the MMs and the benefits of the program:


*“When I met aunty C, she encouraged me … she brought me back to life so I am grateful for that … she is compassionate. Some people will look at you and say it is what you did; that’s the result of your bad character. They would be judgmental. It takes somebody who is compassionate to look at you and say there is still a way out without judging you irrespective of whatever thing you might have done”*
(patient)


*“Some other people would make you feel bad like ‘after all you are positive’. But she carries us all along. She gives us special treatment. That sense of belonging is one thing I have enjoyed so far”*
(patient)


*“She is so lovely. I feel loved … she gives me hope … mentor mothers are patient. They have this type of patience that I cannot describe. Even when you are crying for nothing they keep calming you down”*
(patient)


*“I am so loved. I don’t even know what to say”*
(patient)

The healthcare providers also acknowledged the usefulness and success of the MMs program in promoting and cultivating effective PMTCT care and services:


*“The mentor mothers have the confidence, they can come out and say ‘yes, I am one of you and if you follow PMTCT guidelines, you will stay alive, you can live a positive life’. And people are really motivated by that … The Mentor Mothers program is good and should be sustained because the mentor mothers are doing a great job”*
(PMTCT focal nurse)


*“The patients would describe it as interesting because it has helped them gain confidence in themselves, in their marriages, and in life generally. Like there is this hope, you see. Suddenly the clients will move from being sad and unhappy to being interested and lively once again”*
(PMTCT focal nurse)

### 5.1. HIV Education and Awareness

The findings of this study also revealed that MMs were perceived as a key educational resource for raising awareness and educating women about HIV. Consequently, most of the participants considered the MMs program as an important channel of information and source of insight about HIV. Their experiences were reported as having access to the right information, educating mothers, acquiring knowledge, sensitizing patients, raising community awareness, and empowering patients. Therefore, by taking advantage of the MMs’ community involvement and expanding educational platforms, the Nigerian DoD MMs program focuses on raising awareness of HIV and reducing its prevalence and social stigma.


*“One of the primary things that we do is counselling. This counselling cuts across every aspect. The education of the patient about how the disease can be contracted, the prevention, the complications if they do not take their drug, and the side effects of the drugs”*
(medical officer)

The findings from this study highlight that HIV-positive women who have used healthcare services to prevent vertical transmission of HIV have first-hand experience, which is instrumental to the facilitation and implementation of PMTCT in the Nigerian DoD health facilities. This places them in a position to better educate other HIV-positive women, acting as experienced peer “mentors” for women accessing PMTCT services and their families by providing emotional support, education, and advice [27]. These results echo that the core of the MMs program is access to antiretroviral therapy (ART) with positive outcomes for both mother and baby. To achieve this, trained MMs and other staff within peer group settings provide individual support for HIV-positive pregnant women and postpartum mothers to help them address unmet needs for understanding HIV, psychosocial support and acceptance, self-care, infant care, and, over the longer term, economic needs [28].


*“We counsel them to the point that they are interested in helping others. They reach out to other positive mothers and talk with them, educate them, and give them enough knowledge for them to understand that it is possible to have an HIV-free baby even when the mother is positive”*
(medical officer)


*“We give them adherence counselling right from the day they are declared positive. This is followed by home visits. We visit them to know how they are taking their drugs … we follow them up and based on that they are doing very well”*
(PMTCT focal nurse)

### 5.2. Exclusive Breastfeeding and Mitigation of Vertical Transmission: Outcomes of the Program

MMs were perceived by all the participants as a valuable asset in promoting exclusive breastfeeding, mitigating vertical transmission, and promoting healthy living among HIV-positive women. PMTCT patients within the Nigerian DoD MMs program are advised to breastfeed exclusively for six months and adhere to treatments and all the protocols required. The MMs and staff reinforce this advice and follow up feeding practices and treatment in the PMTCT patients’ homes. The findings from this study revealed that PMTCT patients were in general more positive toward early antenatal check-ups, and all the participants associated these positive outcomes to the influence of the MMs. The findings further revealed that MMs encouraged patients to adhere to their treatment plan and healthcare, test their children after delivery, return to the health facilities for their children’s test results, and adhere to postpartum PMTCT protocols. Thus, instilling hope in PMTCT patients by MMs was just as significant as giving practical advice on daily issues, which promotes or results in positive outcomes.

This was corroborated by these participants:


*“…formally our children were turning out positive … before I became the focal person here there were lots of positive babies, but these days, hardly will you see any positive baby. Since this mentor mother came, we have not recorded any positive baby from our unit. Whenever patients come to the clinic, they always appreciate us, they appreciate the mentor mother in our facility”*
(PMTCT focal nurse)


*“PMTCT is functioning very well. Recently, people have the confidence, the belief that when they are adherent on their medication they are most nearly absolutely sure they are going to have a HIV negative baby. The tracking is reduced, they all just come by themselves. Maximum is a day or two they miss their appointment”*
(medical officer)


*“This program is okay… Mentor Mothers and PMTCT…. great job. I want the program to continue so we can achieve that zero transmission level”*
(PMTCT focal nurse)


*“The program has helped mothers live well … most mothers feel they will not give birth again…but with education by this mentor mother to live healthy, you can give birth to as many children as you like. Provided you keep to the rules, your children will come out healthy. The contribution of the mentor mothers to this program is very good”*
(patient)

### 5.3. Promoting Healthy Patients and Healthcare Relations

Distrust toward healthcare workers and treatment was common amongst most patients in the Nigerian DoD MMs program. Despite knowing that the treatments are reliable, many of the patients were at first reluctant to follow the protocols and instructions given to them. At the time of the study, the facilities’ ART pharmacies were separated from the rest of the hospital pharmacies, with most clinics having separate days or sections for HIV-positive patients for refilling and counselling, which most patients perceived as a reason for lack of confidentiality. However, with the help of MMs sharing their personal experiences with patients, they served as a link between the patients and the healthcare system. As a result, the MMs program in the Nigerian DoD functions as a bridge between patients and the healthcare system, thereby promoting a healthy environment for patients to have access to quality healthcare services in the facilities. It also fills the gap between patients in the community and healthcare services and focuses on HIV awareness and stigma reduction. In other words, the program focuses on advocacy, promoting patients’ rights, teamwork, healthy work relationships, and promoting access to quality care. Hence, most participants perceived MMs’ roles not only as filling the gaps that the healthcare system and the community could not fill but also as educating women on nutrition and feeding practices, facilitating healthy relationships, and providing referrals within the facilities when necessary. In addition, the MMs were perceived as a source of hope and encouragement by helping participants cope with their HIV status, something that some team members could barely provide for HIV-positive women. 

One of the MMs cited such an instance:


*“Even though they have been trained, some still have misconceptions about HIV. Some of them still believe that the child has traces of HIV, so in those kinds of doubt, they fall back to us, we that are positive and have had negative children”*
(mentor mother)

Another MM substantiated the above:


*“I have been able to take away fear from our women by encouraging them. Some left school due to fear. I protect my women … many families depend on us as mentor mothers”*
(mentor mother)

Again another MM validated the above narrative:


*“Because we have a good rapport with them, they trust us … we protect them in the various sites so they come to clinic with boldness knowing the mentor is there … No mentor mother, no PMTCT. It’s just that simple”*
(mentor mother)

From the above narratives, it is implied that there is an expectation for professionalism within the organization to be maintained in order to build trust and good patient–provider relationship. Such a professional commitment was also verified by most of the participants as reported below: 


*“Follow up, good result, encouraging them, and making sure that the confidentiality of these patients is maintained … those are the things that are encouraging the patients to fulfil good interpersonal relationship”*
(medical officer)


*“Those that access care here are very comfortable because they can walk in at any time and meet a particular set of persons and there is no breach in confidentiality”*
(medical officer)


*“They look out for the way you receive the patients, whether you maintain confidentiality or attend to them well”*
(PMTCT focal nurse)

## 6. Discussion

The findings of this study suggest that all participants credit the role of MMs as being effective and instrumental in achieving the goals of the MMs program within the Nigerian DoD PMTCT care and services. These findings not only portray MMs and the program as a key educational resource for educating women about HIV but also reveal that antenatal patients in MMs programs are more likely to return for testing compared to mothers not in the program; among those tested, mothers in a MMs program are more likely to return for their test results. This is largely because of the psychosocial support and HIV awareness that MMs provide to patients. This includes the facilitation of HIV education through community involvement and group health talks, referring clients to PMTCT, HIV care and treatments as needed, maternal and child health support, providing counselling to HIV-positive women and couples, and follow-up with clients who have missed clinic visits for PMTCT. Thus, the findings of this paper highlights that highly structured MMs program further improves uptake and successful completion of PMTCT services within the Nigerian DoD. Similarly, an analysis of mother support groups and MMs programs in countries like South Africa, Kenya, Mozambique, Nigeria, Ethiopia, Malawi, Zimbabwe, and Democratic Republic of the Congo (DRC) have revealed that support groups and MMs play a key role in ensuring a smooth continuum of care for HIV-positive mothers and their children, with MMs programs and/or mother support group participants having better PMTCT outcomes and reporting psychosocial wellbeing more often compared to others [17,20,29,30,31,32,33,34,35,36]. The findings of this study also reveal that structured MMs program impact on other outcomes, such as facility deliveries, new infant infections, breastfeeding, infant survival, and maternal viral suppression, thereby reducing prevalence. More so, the findings from this study and other studies show that HIV-positive antenatal patients in these groups are significantly more likely to reveal their HIV status, undergo CD4 tests during pregnancy, receive ART and practice adherence to care for themselves and their infants, and practice an exclusive infant feeding method. This study’s findings echo the findings of some quantitative studies conducted in Zimbabwe that revealed that retention rates among mothers in a MMs program were higher (99% versus 50%) and that these mothers were more likely to follow PMTCT protocols than those who were not in a mother-to-mother program [37,38]. Therefore, the present analysis of healthcare workers, mentor mothers, and patients’ experiences of the Nigerian DoD MMs program revealed that women in the MMs program report more self-efficacy and motivation, improved retention, empowerment, and increased disclosure of their HIV status.

## 7. Conclusions

The results of our study suggest a high level of satisfaction and positive outcomes from the MMs program orchestrated by the Nigerian DoD for facilitating PMTCT care and services. The participants considered the MMs program as an important medium of information and source of knowledge about HIV. They reported that access to the right information and education adequately equipped them with the right knowledge and sensitization to raise community awareness in the prevention of pediatric HIV infections. Therefore, the present evaluation of healthcare workers, mentor mothers, and patients’ experiences of the Nigerian DoD MMs program revealed that women in the MMs program report more self-efficacy and motivation, improved retention, empowerment, and increased disclosure of their HIV status, which is a positive sign that the program is achieving its goals. 

### 7.1. Limitations of the Study

The following were limitations of this study:Due to the regimented nature of the Nigerian military, very limited information exists in the public domain on prevention of mother-to-child transmission of HIV using mentor mothers in the Nigerian DoD. In addition, research of this nature has not been conducted before now in the DoD, so researchers had limited literature on this subject in the DoD.Very few studies have previously addressed the experiences of healthcare workers, mentor mothers, and patients. The unavailability of adequate literature therefore limited extensive presentation of the discussion of findings in this study. The study was limited to Abuja, the Federal Capital City of Nigeria, which was chosen because it is the administrative headquarter of the Nigerian Military where the MMs program was implemented in 2014. However, due to the specific number of participants in this study, coupled with the fact that the study was conducted in one region of the country, the study findings cannot be generalized. It is suggested that further research be conducted in other DoD facilities to further explore the MMs strategy for PMTCT.

### 7.2. Recommendations

Further research should also be conducted on the impact of the MMs program in the community. In addition, community awareness of MMs programs for PMTCT and its effectiveness should be investigated to measure the knowledge of women and mothers of childbearing age about PMTCT practices. This can provide insight on which areas and population groups to direct PMTCT care and services.

## Figures and Tables

**Figure 1 healthcare-09-00328-f001:**
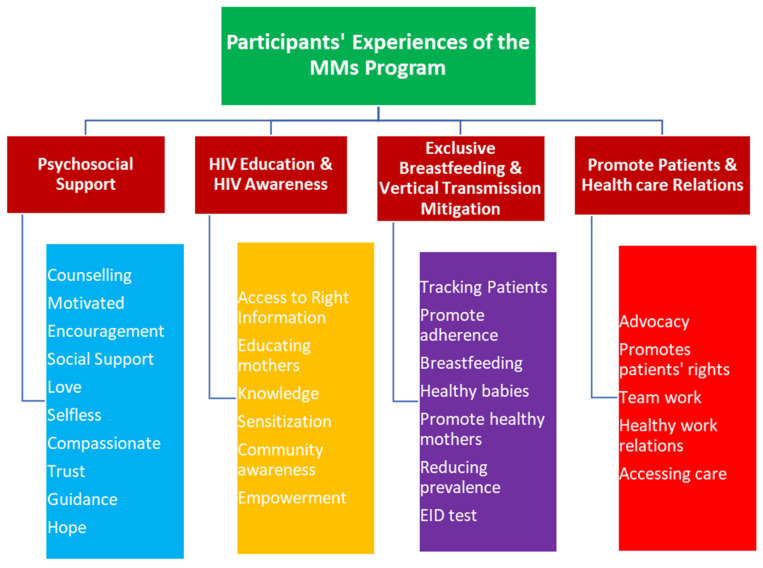
Summary of participants’ experiences of the Nigerian Department of Defense (DoD) Mentor Mothers (MMs) program.

**Table 1 healthcare-09-00328-t001:** Participants profile.

Pseudonym	Professional Position in Unit	Duration of employment
Effa	Medical officer (PMTCT)	4 years
Kalan	Medical officer (PMTCT)	13 years
Ovede	Medical officer (PMTCT)	12 years
Fenky	Nurse (PMTCT focal person)	16 years
Mokun	Nurse (PMTCT focal person)	22 years
Ntewa	Nurse (PMTCT focal person)	20 years
Cyuko	Mentor Mother	3 years
Dobla	Mentor Mother	4 years
Nancel	Mentor Mother	2 years
Neki	Mentor Mother	5 years
Nelly	Mentor Mother	5 years
Savic	Mentor Mother	5 years
Cecio	Patient	
Chima	Patient	
Chuke	Patient	
Hilda	Patient	
Katu	Patient	
Mabel	Patient	

## Data Availability

This is a qualitative study and the participants did not consent to having their full transcripts made publicly available. However, data on this study maybe made available upon reasonable request to the relevant stakeholders and the authors: mbuba4real@yahoo.com; mhlongoem@ukzn.ac.za.

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
