# Peer review of "The Mentor Mothers Program in the Department of Defense in Nigeria: An Evaluation of Healthcare Workers, Mentor Mothers, and Patients’ Experiences"

_healthcare, 2021, doi:10.3390/healthcare9030328_

Round 1

Reviewer 1 Report

The authors report on a qualitative study of interviews with healthcare workers, Mentor Mothers, and patients to understand the experiences of the Mentor Mothers program in Nigeria. The authors identified 4 main themes using thematic analysis. My comments are arranged into major and minor considerations:

Major:

  • The background section provides a nice review of the problem, but I got lost in understanding if the main point was about children, pregnant women, or mother-to-child transmission as some paragraphs addressed all 3 populations? It would be helpful to streamline the introduction to be clear in the population or populations of interest. The aim of the paper states that you are targeting PMTCT, but the background does not make that clear.
  • I recommend not including the participant profile as this could possibly identify the participant. Even though the authors use pseudonyms, the professional position, employment duration and knowing what locations were used for the study could potentially identify the subject. I would delete table 1, and in the results just state years of employment for each participant. Also in the results after each example sentence, instead of pseudonym, just put their role. This would increase anonymity.  
  • Data analysis:
    • Please clarify if it was just the single author who conducted the interviews.
    • More detail is needed in the data analysis section. Describe how codes and themes were formed. The authors state that they identified common constructions or meanings, but it’s not clear if these were the codes, or how that was organized.
    • Was it only 1 person who read and determined codes and themes? To increase trustworthiness, it is recommended that the data is reviewed by more than one person independently, and codes are discussed with at least 1 other person using an iterative process to determine final codes/themes.
    • Were any of the findings checked with the participants? It would be beneficial if the themes were reviewed with some of the participants to see if the findings captured the experience of the participants.
    • Can you provide the questions asked in a supplement table? Were the same questions asked to all participants? A few example questions at least would be helpful.
  • The overall discussion is lacking. There should be more contextualizing of the qualitative results within the mentor mothers program, or how it fits within the literature. There were no limitations discussed.

Minor:

  • 1st paragraph, define UNAIDS, HIV, ARV
  • You use the abbreviation for mentor mothers (MM), but then continue to spell out mentor mothers for much of the rest of the paper. Recommend being consistent with either MM or spelling out mentor mothers.
  • Personally, I do not think it’s necessary to state that this study was part of a PhD research training, or that it’s part of 4 papers (as stated in the aims). I think you should just reference the others if they are published, or just not include this statement. It doesn’t add anything to the paper to include this information.
  • This was not a case study. Recommend removing the word “case” from the research design section.

Reviewer 2 Report

The article by Ibu and Mhlongo presents an interesting investigation of stakeholder perspectives on a mentor support program to promote engagement in PMTCT care in Nigeria. The purpose and methods are clear and acceptable, though in the results section the presentation of themes and corresponding participants quotes reflecting these themes are mixed together (overlap among themes, quotes including in what appear to be the incorrect theme). Moreover, the discussion is not appropriate for the data presented in the study – focusing on results related to effectiveness of the program in promoting PMTCT engagement rather than a discussion of the impact of the program on the outcomes discussed in results. While the data are strong enough to warrant a paper, the manuscript requires substantial revision to address the unclear results presentation and inadequate discussion. Below I describe specific areas that require revision.

Abstract

“Some selected hospitals” should give more information about number, characteristics, motive for selection

Should include what method was used to analyze qualitative data (thematic analysis).

Seems odd to describe these quantitative type results about project effectiveness

Introduction

It may seem unnecessary, but I suggest the authors write out what the ARV acronym stands for the first time it is used.

Methods

Research setting needs to be described with more detail. “Some” should be replaced by a specific number of health facilities. Also, it is unclear by the description of the setting if all participants are military-related or if these facilities also provide care to the general population (patients who are non-military affiliated).

It is unclear if the participants have all been in the program for over 6 month or just providers and MM. The sentence describing this should explicitly state if this is just referred to providers and MM, and patients were selected by participation in at least 4 clinic visits.

The description of the instrument should provide data on what types of questions were included in the guide.

Table 1. It would be interesting to include a column showing how long each participant had been part of the MM program.

Results

Figure 1 is not referred to anywhere in the text. I believe it should be included in the first paragraph of findings. Also the figure requires a legend to understand what the words within each theme are meant to represent.

It would be interesting to include a quote from the patients about HIV awareness and education. The current quotes from MM and providers do not represent whether the patients also felt the program was an effective way to promote awareness.

The part on “hope” within the section on “promoting healthy patients” seems very out of place, particularly because hope and encouragement are previously described in the psychosocial support theme. Additionally, the first quote very clearly seems about HIV education, and should be incorporated in the section on that theme. The second quote also seems out of place as it describes psychososical support (encouragement, taking away fear), not health promotion, and thus should be moved to the psychosocial section.

Either the order of Figure 1 themes or the order of the results presentation should be changed so that they are aligned. The text has swapped the order of Exclusive breastfeeding and Promoting healthcare.

Discussion and Conclusions

“These findings not only portrayed mentor mothers and the program as a key educational resource for educating women about HIV, it also revealed that antenatal patients in Mentor Mothers programs were more likely to return for testing as compared with mothers not in the program; and among those who were tested, mothers in a Mentor Mothers program were more likely to return for their test results.”

“More so, the findings from this study and other studies further revealed that HIV-positive antenatal patients in these groups were significantly more likely to reveal their HIV status, undergo CD4 tests during pregnancy, receive ART and practice adherence to care for themselves and their infants, and practice an exclusive infant feeding method”

“Therefore, the analysis of the health care workers, mentor mothers, and the patients’ experiences of the Nigeria DoD Mentor Mothers program revealed that women in the MMs program reported more self-efficacy and motivation, improved retention, empowerment, and increased disclosure of their HIV status.”

“Therefore, an evaluation of the health care workers, mentor mothers, and the patients’ experiences of the Nigeria DoD MMs program revealed that women in the MMs program reported more self-efficacy and motivation, improved retention, empowerment, and increased disclosure of their HIV status which was a positive sign that the program is achieving its goals.”

The above findings stated in the discussion are not demonstrated in the paper. These descriptions would be fitting of a quantitative study that included both cases and controls. The present study only included qualitative data and participants who had taken part in the MM intervention. The discussion needs to be rewritten to discuss the qualitative findings, and linking them to previous research on qualitative investigations of similar programs. While the results of the present study are suggestive that the program promotes PMTCT, it does not actually demonstrate that.

Further, a imitations section is required that describes, for example, the limitation of having what appears to be only 1 person analyzing the data (standard would be at least 2 for qualitative analysis) and only including women from one region (which is perhaps not representative of Nigeria as a whole).

Author Response

All 

Round 2

Reviewer 1 Report

Reviewer's comments adequately addressed. General editing for grammar and word flow still needed.

Considering that there will be English editing provided, the paper can be accepted in its' current form.